# Nutritional Assessment of Baby Food Available in Italy

**DOI:** 10.3390/nu14183722

**Published:** 2022-09-09

**Authors:** Angelo Antignani, Ruggiero Francavilla, Andrea Vania, Lucia Leonardi, Cristina Di Mauro, Giovanna Tezza, Fernanda Cristofori, Vanessa Nadia Dargenio, Immacolata Scotese, Filomena Palma, Margherita Caroli

**Affiliations:** 1Department of Food Science, University of Naples Federico II, 80100 Naples, Italy; 2Interdisciplinary Department of Medicine, Pediatric Section, Children’s Hospital ‘Giovanni XXIII’, University of Bari Aldo Moro, 70126 Bari, Italy; 3Independent Researcher, 00162 Rome, Italy; 4Maternal Infantile and Urological Sciences Department, Sapienza University, 00161 Rome, Italy; 5Regional Centre of Pharmacovigilance Campania Region, Department of Experimental Medicine, University Luigi Vanvitelli, 80138 Naples, Italy; 6Department of Pediatrics, San Bortolo Hospital, 36100 Vicenza, Italy; 7Health District 64, ASL Salerno, 84022 Campagna, Italy; 8Health District 65, ASL Salerno, 84091 Battipaglia, Italy; 9Independent Researcher, Francavilla Fontana, 72021 Brindisi, Italy

**Keywords:** complementary feeding, weaning, commercial baby food, nutritional composition, macronutrients, fiber, micronutrients

## Abstract

Adequate complementary feeding practices are important for short- and long-term child health. In industrialized countries, the formulation of several commercial baby foods (CBFs) and an increase in their consumption has been noticed. Aim: To update and analyze the nutritional composition of CBFs available in the Italian market. Methods: Data collection carried out in two steps (July 2018–January 2019) and updated in May–September 2021. The information on CBFs was taken from the websites of the major CBF producers available in Italy. The collected information were: Suggested initial and final age of consumption; Ingredients; Energy value; Macronutrients (protein, lipids, and carbohydrates); Fiber; Micronutrients (sodium, iron, and calcium); Presence of salt and added sugars, flavorings, and other additives. Results: Time-space for which CBFs are recommended starts too early and ends too late; protein content is adequate and even too high in some food; Amount of fats and their quality must be improved, keeping the intake of saturated fats low; Sugar content is too high in too many CBFs and salt is unnecessarily present in some of them. Finally, the texture of too many products is purée, and its use is recommended for too long, hindering the development of infants’ chewing abilities.

## 1. Introduction

It is well known that feeding in the first thousand days of life has enduring effects for the rest of life. The complementary feeding (CF) period falls within a critical time when an environmental factor, such as feeding, can alter infant development with a long-term effect even into adulthood [1]. 

WHO recommends that infants should be exclusively breastfed for the first 6 months of life and thereafter should receive nutritionally adequate and safe complementary foods while breastfeeding continues for up to two years of age or beyond [2]. 

Foods offered to the infant during the CF period depend on the availability, local culture, the advice of the pediatrician and other health care professionals, but also on the beliefs and economic possibility of the family, and finally, on the effect of research done independently by parents on the web, which is full of websites and blogs ready to give various advices for this age group, although they are not always adequate [3].

An initial choice that families make is between foods prepared at home and foods prepared by the industry, especially for children under thirty-six months of age. In industrialized countries, the increase in the number of working mothers, and new lifestyles in general, have resulted in the formulation of several different types of commercial baby foods (CBFs) and an increase in their consumption [4,5,6].

In the European Union, CBF production is regulated both by European laws [7,8] and by the national regulations. Specifically, in Italy, CBFs are generally obtained from controlled supply chains, often from organic crops. Moreover, the safety requirements prescribed by Italian regulations on the presence of contaminants and pollutants in preparations for infants and young children are much more restrictive than those applied to the “organic” system, so the words “organic product” on CBF labels do not add additional safety measures. CBF, therefore, has a high guarantee of quality and safety of the production chain and a lower risk of contamination, as regulated by special laws of the Italian Republic [9].

While a wide variety of products can help families to best choose the foods that suit the infant’s diet and to the specific needs of the family, the same wide variety can confuse the family and encourage bad choices and advertise products that are unnecessary (and/or even potentially harmful because of unhealthy nutritional composition) to the infant’s nutritional status and his/her future feeding habits. Indeed, eating patterns and food preferences start from an early age; thus, the first three years of life can be considered an important window to influence a life-long healthy eating behavior [10]. 

### Aims

To our knowledge, there is no updated and complete database on CBFs and their nutritional composition available in Italy. A precise comparison between their content of nutrients and the recommendations of the WHO on the correct composition of the different CBFs has never been performed. The purpose of this study was to explore the CBFs present in Italy and to verify whether the nutritional composition is adequate for the needs of infants, the proper development of taste and thus eating habits at a later age, and finally, whether the products comply with the WHO recommendations in this regard.

## 2. Materials and Methods

The first data collection was carried out from July 2018 to January 2019 upon the request of the WHO to one of the authors (M.C.) to participate in a European study on the availability and characteristics of baby food in Europe [11].

The 2019 data collection was conducted by only one of the authors (A.A.); the second collection was made by six of the authors (A.A., C.DM., I.S., F.C., F.S., F.P). Each of them checked the products of a company and compared them with those of 2019. After the comparison of the two lists, four products were removed because they were no longer on the market. In June 2022, the six authors in charge of the research once again visited the websites and found the notable increase in the number of CBFs based on legumes. Consequently, the group of legumes has been updated. The information of the manufacturers that are available in the market nationwide and their products’ nutritional data were collected mainly from the manufacturers’ websites. All categories of food products intended for infants and young children from four to thirty-six months were collected and then analyzed; milk replacement formulas were not included as they are not really considered as “complementary food” in general opinion, although they are considered as such by the WHO. The broth products were not included because their nutritional content is both very low and extremely dependent on the amount of water used for the preparation. When out-of-scale nutritional values were found on the web site, the values were checked on the product’s label on sale in supermarket, pharmacies, drugstores or other sale places. The collected information were: (a) suggested age for starting intake and suggested final age, if any; (b) ingredients as reported in the product’s label; (c) energy value; (d) macronutrients (protein, lipids, and carbohydrates); (e) fiber; (f) some relevant micronutrients (sodium, iron, and calcium); (g) presence of added salt and sugars, and (h) presence of flavorings and other additives.

The data were classified in agreement to the WHO classification [12]. Briefly, the gathered data were collected in an Excel spreadsheet, in which products were divided into categories based on the ingredients (fruits, vegetables, legumes, dairy products, meat, fish, and grains) and the type of production (sauces, freeze-dried, baby food cereals, and snacks). Products in which vegetables and/or tubers were also present in substantial addition to meat or fish protein were classified in a separate category (“meat or fish purée with vegetables”). Subgroups were then created to categorize the products more in line with Italian eating habits.

For each category, the following information were entered in the database: (a) name of the product; (b) the recommended age for consumption (from–to); (c) ingredients as reported on the product’s label; (d) total energy/100 g or 100 mL (expressed in kcal); (e) protein (g); (f) total fat (g); (g) saturated fat (g); (h) carbohydrates (g); (i) simple sugars (g); (l) fiber (g); (m) sodium (mg); (n) iron (mg), and (o) calcium (mg). For meat products, where the nutritional label was not available, the iron values were calculated according to the individual animal protein source using the food composition tables of the Council for Agricultural Research and Analysis of Agricultural Economics [13]. The amount of iron actually present in a specific product was assessed by calculating the proportion of meat contained in the food under analysis and relating it to the reference tables.

Added sugars were identified according to the 2010 definition of the European Food Safety Authority [14], also including fruit juices (and derivatives) and honey as established by the WHO in 2015 [15]. Among added sugars, neither lactose was included, as it is present as a component of milk and/or milk products, nor sugars coming from fruit purées, where their composition was not specified in detail. The presence of added sugars, although reported in the ingredients in their several type and definitions, is not stated in labeled amounts, and therefore it was not possible to calculate their amount and percentage over total energy intake, but the amount of total sugars present was calculated.

### Statistical Analysis

Mean and standard deviation calculations were made for each category using the respective Excel functions. In addition, mean and standard deviation of the subgroups in which some specific added ingredient was present were also calculated for each category.

## 3. Results

A total of two hundred eighty-five products were found. The least numerous product group was “early teething meat”, with only four products from a single company. The most significant product group was “snacks” with one-hundred-twenty-five products; within snacks, the subgroup “spoon baby snacks” with seventy-three products. The second most represented group was “cereal baby food” with sixty-four products (thirty-nine types of baby pasta and/or noodles and twenty-five different types of dry cereals), followed by “fruit purée” with thirty-nine products, almost tied with the subgroup of “drinking snacks” (thirty-seven). One hundred forty-five products (37.1%) were recommended from four months of age: 100% of the freeze-dried meat CBFs, 79% of the baby fruit purées, and 36% of the cereal CBFs. Ninety products (23%), distributed among the various groups, recommended thirty-six months as the final age of consumption. The recommended starting and ending age timings are shown in Table 1. The results of the nutritional analysis of the products are shown in Table 2.

### 3.1. Fruit Purée Baby Foods

Of the thirty-nine fruit purée products on the market, thirty-one are recommended from the age of four months, and only eight are recommended from six months; eight products give thirty-six months as the last age for consumption, while the others do not suggest a final period of consumption. Twenty-four products come in a single 100 mL package, nine in 80 mL packages, three in 125 mL packages, two in 80, 100, and 120 mL packages, and one in 100 and 120 mL packages; all have the same nutritional composition. Sixteen out of thirty-nine (41%) products contain added sugars. Only nine products are based on a single type of fruit. All others are composed of mixed fruit in which the most often represented ingredient is apple (twenty-six products), followed by banana (fourteen products). No products are fortified with calcium or iron; the sodium content is the one naturally found in fruit.

### 3.2. Cheese Purée Baby Foods

There are thirteen products available: eleven have a recommended start at four and two at six months. All packages are 80 g. Three products list thirty-six months as the maximum age for consumption. Three products, besides dairy, report on the label the presence of cooked ham as an ingredient. For eleven products, the first ingredient is water; five products contain rice starch, four corn starch, and four more contain both ingredients. The percentage of dairy contained in the products is about 40%; for the three CBFs containing also ham, the dairy percentage drops to 23%. The average protein content is very high (8.7 ± 2.7 g/100 g of product) even considering the consumption of half a jar (40 g = 3.5 g of proteins). Overall, the average sodium contained (235.7 ± 130 mg/100 g of product) is related to sodium cheese content and is similar to the average sodium contained in the products with added sodium (240.0 ± 146.6 mg/100 g of product). Although they are milk-derived products, six of them do not declare the amount of calcium content on the label. When declared on the labels, calcium content ranges from 65 mg/100 g to 430 mg/100 g of product. Only two products are fortified with calcium, but the amount is not specified and the total amount of this micronutrient is less than in non-enriched products. 

### 3.3. Meat-Only Purée Baby Foods

There are thirty products in this group, twenty of which state four months as the recommended starting age, while ten recommend starting at six months. Only seven products mark the last feeding age, at thirty-six months. Twenty-one products have only the 80 g package; among the products recommended starting at four months, nine are on the market with one 80 g package and one 120 g package. The percentage of meat from different animals is always 30% of the total; therefore, the average protein content is 6.3 ± 0.5 g/100 g of product. All have flour or rice starch among the ingredients. Only seven products declare the presence of sunflower oil, so the fats in the other formulations are supposed to come only from the lipids of the meats. Only two products declare the presence of added salt; in both cases, it derives from the cooked ham used as a protein source, but the sodium content in all products is low (36.6 ± 32.2 mg/100 g of product). No products are enriched in iron or calcium. 

### 3.4. Meat or Fish Purées with Vegetables

There are twenty-three products, nine of which suggest as the starting age four months and fourteen suggest six months of life; only five products report the maximum age for consumption (thirty-six months). Only 80 g packages exist for this category. In five products, the protein source is meat from different animals (25% of the total product’s weight), while in the remaining eighteen CBFs, the source is fish of varying types. The average percentage of fish is 18%, and only five products reach 20%. None have added salt, except for one product made from cooked ham in which iodized salt is a constitutive ingredient. Protein content averages 3.9 g ± 0.7 g/100 g of product. Data on iron content are available for a few products, but in any case, the small amount of meat or fish present makes this information superfluous.

### 3.5. Vegetable Purées

There are twelve CBFs made of vegetable purée on the market. Eleven are recommended from four months and only one from six months of age. Three products have thirty-six months of life as their final recommended age. All are in 80 g packages. No product has added sugar or salt. Four products contain only one type of vegetable (two are made of carrots, one of broccoli, and one of squash). All the others consist of mixed vegetables. The vegetables most frequently found in these products are carrots, followed by zucchini. Nine products contain potatoes listed among vegetables, although they are tubers, and five include spinach. Only two products include bitter vegetables, one broccoli alone, and one cauliflower with other vegetables. All products are gluten-free, but only eight state this point on the label; none claim to have added salt, but, strangely, two products both containing carrots, one with an 80% and the other one with a 90% concentration of this vegetable, have very different contents: the first contains 36 mg of sodium and the second 80 mg (per 100 g of product).

### 3.6. Legumes Purées

There are fourteen legumes purée baby foods on the market. For five products, the consumption is recommended from four months of age, for six products from six months, and for the last three from eight months; no product reports a maximum age for consumption. Five products are based on a single legume (lentils, beans, chickpeas, peas), while nine contain multiple different legumes and/or added vegetables and/or rice flour. No product has added sugar, but three have water as the first ingredient and a legume percentage of about 20%. According to the list of ingredients no products have added salt, but this is only stated on the label of five products. No products have added iron and/or calcium, and the amount naturally coming from the legumes is not displayed. 

### 3.7. Early Teething Meat

Only one company markets this type of CBF, early teething meat, made of lumped meat, with four different products, for which the company recommends a starting age of eight months and an ending age of thirty-six months; the packages are all 80 g with 30% meat per 100 g of product, with no added salt. The protein content is 6.2 g per 100 g of product in all the products. The nutritional composition is similar among the different preparations, the only nutrient with appreciable differences being iron, the amount of which depends on the type of meat used (0.44 ± 0.26 mg/100 g of product). None of these products is fortified with iron or calcium, and the ingredients’ sodium content is only that naturally contained in the meat (25.6 ± 2.3 mg/100 g).

### 3.8. Tomato Sauces

There are nine products in this group: six recommend eight months, while three recommend ten months as the starting age for consumption. Only three products indicate the maximum period of consumption at thirty-six months. The majority are tomato sauces with ricotta or mozzarella or beef. In detail, four have meat among the ingredients, and three have dairy products; sauces with these ingredients have a higher protein content than those prepared only with tomatoes and vegetables (3.6 g vs. 1.2 g/100 g of product). Six products have added salt and one also sugar. No product has cereals with gluten among the ingredients, but “gluten-free” appears only on the label of 3 out of 9 products.

### 3.9. Freeze-Dried Meat Products

The freeze-dried meat products group has eleven products with a recommended starting age of four months; only three products are advised to stop after thirty-six months. No product has added salt. All packages are 10 g, and 100 g contain the equivalent of 220/240 g of fresh meat. Each pack contains an average of 4.8 ± 0.3 g of protein. No product has cereals with gluten among the ingredients: all have rice flour, and five products also have cornstarch. The statement “gluten-free” appears only on the label of three products. Oil does not appear among the ingredients; therefore, the fat present is supposed to derive exclusively from the ingredients used. 

### 3.10. Milky-Cereal Baby Foods

The milky-cereal group comprises milk together with cereal flours (rice, corn) with the addition of sugar, vitamins, and minerals. There are thirteen products, two of which report four months as the starting age and eleven report six months. Only two products report thirty-six months as the last age of consumption. Total sugars represent 32% of the caloric intake; eleven products contain added sugars. All products are fortified with vitamins and iron, and twelve also with calcium. Ten products contain gluten, and seven include fruit. “Natural flavors” and vanillin are present in the ingredient list of three products each. According to the Italian regulation, all products do not contain dyes and preservatives, as they are not allowed. 

### 3.11. Complete Meals Baby Foods

There are nine products in the complete meals group; all recommend six months as the starting age, while only six recommend stopping consumption at thirty-six months. Two products have added salt; two are enriched with iron and calcium, although the formulation and the amount of both minerals are not specified; eight have meat as an ingredient (8% of the total weight), and one has cheese (14% of the total weight). All have vegetables among the ingredients. These products only require heating without adding other ingredients.

### 3.12. Dry Cereal Baby Foods

A total of sixty-four products composes the dry cereals group. For the analysis, the group was divided into (a) dry cereals to be prepared with hot water or vegetable broth and (b) baby pasta to be cooked in boiling water according to Italian tradition. 

There are twenty-five dry grounded cereal products, eighteen of which are recommended from four months and seven from six months. Only four products (16%) suggest thirty-six months as the age limit for consumption. All products are fortified with vitamins, particularly vitamin B1. Three products are fortified with iron and calcium and four with calcium only; the amounts are specified. One product is added with vanillin. Six are declared gluten-free; four use only naturally gluten-free cereals in their composition, but do not specifically state “does not contain gluten” on the label. Eleven products communicate the use of gluten-containing grains in the recipe; three do not declare it on the label.

The thirty-nine types of baby pasta differ mainly in the shape and size of the dough. Six products are recommended from four months, eight from five months, twelve from six months, five from eight months and eight from ten months of age; fourteen products (35.9%) give the final consumption age at thirty months. The different starting ages depend on the pasta shape and size as well as the ability of the infant to handle this type of food. Twelve products are fortified with calcium and iron, and the amounts are specified. 

### 3.13. Biscuits

There are nineteen products in this group. Six of these products are granular biscuits to be added to the milk; only three are gluten-free. Seven are recommended from four months, six from six months, one from ten months, and finally two from twelve months of age. Only three products suggest stopping the use at the age of thirty-six months. All have added sugar, and five also have salt. 

### 3.14. Baby Snacks

This group is the most abundant with products of various kinds. They can be milk, yogurt, or fruit-based to eat with a spoon or drink. Seven baby snacks belong to the subgroup of bite-sized snacks and consist of galettes or snacks with soft cookies and milk cream filling.

#### 3.14.1. Spoon Baby Snacks

There are seventy-three spoon baby snack products, five of which are recommended as early as four months, eleven from the age of five months, forty-five from six months, three from eight months and ten months, and two from twelve months. Only nineteen (26%) products report thirty-six months as the last age for consumption. Sixty-seven products (91.7%) have added sugars (49% of total energy), and one has added salt. Five products are fortified with iron and four with calcium, but the amount is not specified. Thirty products have fruit as the top-ranking ingredient of the product on the label, thirty-five use the word milk or yogurt in the name, two are labeled as pudding, two as junior desserts, and two as “creamy snacks”. Forty-six products have whole or semi-skimmed milk or yogurt as the first ingredient, fourteen have fruit, and twelve have water.

#### 3.14.2. Drinking Baby Snacks (Fruit Juice, Yogurt)

There are thirty-seven products in this category, eleven of which can be proposed from four months, twenty from six months, one from eight months, two from ten months, and three from twelve months of age. Only four (10.8%) suggest stopping the use at thirty-six months. Overall, the energy derived from sugar is around 68%, while in twenty-eight products with added sugars, it reaches 76% of the total energy intake. Twenty-four products consist of single or mixed fruit, five of fruit plus yogurt, three of yogurt plus fruit, two of yogurt with the cookie, two of yogurt with cocoa, and one of vanilla yogurt or yogurt only.

#### 3.14.3. Baby-Snacks-to-Bite

There are seven baby-snacks-to-bite products, one of which does not specify the starting age of consumption while the remaining six are recommended from eight months of age. Only two (28.6%) suggest stopping consumption at thirty-six months. Five products have added sugars, and one has salt; two of them list on the label the presence of B-vitamins, calcium, and iron, although the amount or chemical formulation is not specified. Two products consist of cookies in packages of 180 g, but neither the weight of the individual cookie, nor the suggested serving portions are set. The remaining products consist of galettes in packets of 30–35 g placed in packages containing multiple boxes. The recommended portion size for a child by the company is one pack. 

#### 3.14.4. Early Teething Snacks

There are eight products on the market, of which three are recommended from four months, one from six months and four from eight months of age. The manufacturer of the product provides no recommendations about the age when to cease the usage. Although they are all referred to as “Early Teething Snacks”, none of the products require chewing, as they all consist of grated fruit. None of the products have added sugars, and all have a 100 mL package.

## 4. Discussion

The data we reported offer a current picture of the world of infant feeding and suggest infant nutrition returns to the hands of the pediatrician who in the face of a wide choice can advise the mother on the best one for her baby.

The analysis of the nutritional assessment of baby foods available in Italy has brought to light some aspects that require careful evaluation aimed at making a better and safer use of the products on the market for infants below the age of three. The first consideration that emerges is that a vast amount of products (one-hundred-five products = 37%) state that they can be used from four months of age, and in our opinion, this limit needs to be raised especially in view of the WHO recommendation to continue with exclusive breastfeeding or formula feeding until six full months [2].

The EU recently asked the EFSA to verify the possibility of changing the age limit for consumption of foods other than breast milk or formulas from four to six months [7]. EFSA’s response was not clear, instead rather confusing, as it stated that “*As long as foods have an age-appropriate texture, are nutritionally appropriate and prepared following good hygiene practices, there is no convincing evidence that at any age investigated in the included studies (<1 to 6 months), the introduction of complementary foods (CFs) is associated with adverse health effects or benefits (except for infants at risk of iron depletion). The Panel concludes that exclusive breastfeeding is nutritionally adequate up to 6 months for the majority of healthy infants born at term from healthy well-nourished mothers*” [16]. 

A similar Solomonic response is also found in the position papers by the American Academy of Pediatrics (AAP) and the European Society of Pediatric Gastroenterology, Hepatology And Nutrition (ESPGHAN) [17,18]. This lack of straightforward recommendations leaves the ground open for a too early initiation of CF, which is completely unnecessary from a nutritional point of view and theoretically responsible for an early end of exclusive breastfeeding (or formula). It cannot be forgotten that any attitude that might encourage the early abandonment of breast milk without nutritional benefit, deprives the infant of a number of functional nutrients that are useful, if or even necessary, for an optimal psycho-physical development. In addition, the reduction of breastfeeding reduces its protective effect against breast cancer for the mother [19]. Unfortunately, these two aspects of outstanding ethical importance have not been stressed enough in the EFSA statement, when it says “*the fact that an infant may be ready from a neurodevelopmental point of view to progress from a liquid to a more diversified diet before six months of age does not imply that there is a need to introduce CFs*” [19]. As a result of this debate, the European legislation on the minimum age limit for the introduction of CF, has not changed. Companies that ethically place the start of consumption of some of their products at six months of age, as opposed to the same type of product that other companies suggest at four months of age and older, are harmed from a commercial point of view, and this inhibits, even more, the companies’ voluntary change of labels toward starting complementary foods at six months of age at the earliest.

The fact that the final age for consumption of baby foods is set at thirty-six months is extremely incorrect. The use of foods up to three years of age still predominantly presented in a puréed or semi-liquid form that can be consumed without chewing does not facilitate the development of this skill, but promotes functional hypotrophy of the masticatory muscles and, consequently, an impaired development of the maxillofacial joint [20,21]. Furthermore, it appears from available studies that the development of chewing skills may play a very important role, not only in the development of the mechanical part of feeding, but also in that of long-term eating habits. In fact, it appears that offering foods no longer in the form of purée, but with the presence of lumps and less soft parts by ten months of age, but still adequate to the infant’s chewing ability, reduces food pickiness up to 7 years of age [22].

### 4.1. Fruit Purées

The thirty-six months recommended as the final age for consumption of fruit purée is definitely excessive. Indeed, as early as eight months, the infant begins to chew, and therefore, already at this age, crushed fruit and/or in small pieces, could be offered. The frequent addition of sugars to commercial fruit purée also makes the taste of fruit much sweeter, and therefore it could then be more difficult for the infant to become accustomed later on to the natural taste of fruit [23]. Finally, the great predominance of apples in the products makes the flavor, in general, monotonous, and does not contribute to broadening the variety of flavors that the infant should learn to appreciate.

### 4.2. Vegetables Purées

Despite being useful in an emergency situation, the fact that there are predominantly mixed vegetable products prevents the infant from getting to know the specific tastes of individual vegetables. In addition, the vegetables present are predominantly sweet-tasting and do not accustom the infant to appreciating bitter or bitterish vegetables, which are often much richer in protective substances and useful minerals [24].

### 4.3. Legumes Purées

Legumes are an excellent source of protein and have a wide variety of tastes. However, the preparation of legumes may be considered by mothers as too time-consuming to be included in a child’s diet on a regular basis. Therefore, CBFs could facilitate the use of legumes by families with the advantage of making it possible to become familiar with the taste of legumes at an early age, facilitating the consumption of this important food group at later ages. The legume-based CBFs available already include most of the legumes commonly consumed in Italy, except fava beans, but a further expansion of the variety of legumes, including legumes more used in other countries of the world, would certainly be useful. The sudden increase in the number of legumes preparations probably reflects the acknowledgement by the companies of the increased appeal of this food group on the general population and, consequently, the opening of new market opportunities.

### 4.4. Baby Cheese Purées

The relatively high sodium content in these products should prompt their careful and limited use, and not earlier than one year of age. Even if the supposed use is to increase the calcium intake, it should be considered that the calcium content is very variable in these products, since the type of cheese as a main ingredient and the percentage of cheese in the product also varies. In addition, as water is the first ingredient in almost all products, they might be less harmful (less salt offered to the child) but also less useful in increasing significantly the calcium intake. The latter, indeed, can also be increased with the use of conventional cheeses, being careful about the amount of sodium present in the more aged ones and the protein content. The protein content (8.67 g/100 g of product) higher than in all other protein-rich CBF, requires cautious use, certainly their avoidance before six months of age, and an even greater caution if the infant is formula-fed.

### 4.5. Meat-Only Purée Baby Foods

Checking on the websites of the manufacturers, we found that some of these products are included in recipes that suggest to use 40 g of product at six months and 80 g at ten months, without differentiating whether the infant is breastfed or formula-fed. However, the double packaging (80–120 g) of meat-only purée may mislead parents into thinking that the 80 g package is appropriate for younger infants and the 120 g package appropriate for the older ones. This may contribute to the infant’s excessive protein intake. The inclusion of a protein-rich product at four months (6.3 ± 0.5 g/100 g) may substantially increase the protein intake at this age, especially if the infant is formula-fed. Moreover, 24 g of meat (as in the 80 g package) provides only 0.41 ± 0.23 mg of iron, which is well below the Population Recommended Intake (PRI) for this age group (11 mg/day), making these products not useful if they are intended to substantially increase the iron intake. 

### 4.6. Meat or Fish Purées with Vegetables

Data on iron content of these products are very partial, mainly due to the lack of data on the content of some fish in the Italian national databases [13] and therefore it is not advisable to use these products to increase iron intake.

### 4.7. Early Teething Meat

The recommended age for starting consumption is correct, and they can be useful in aiding the acquisition of chewing skills [25]. As opposed to this positive aspect, for an 8-month-old infant, the PRI for protein is 11 g; consequently, the consumption of a whole package of early teething meat would provide more than 50% of the daily protein Dietary Recommended Values (DRV). Thus, even in this group, the amount of protein coming from the suggested serving size may be excessive.

### 4.8. Tomato Sauces

The presence of meat or dairy products in some of the sauces in this group increases the protein content by about three times compared to that of a simple tomato sauce and, especially in formula-fed infants, this can increase excessively the protein intake. Presenting these complex sauces, no recommendation is given in the websites about avoiding other protein-rich foods in the same meal.

### 4.9. Freeze-Dried Meat Baby Foods

The age of thirty-six months as the maximum consumption period is excessively high for the freeze-dried meat group. Indeed, the child should already be able to eat food that differs in taste and texture every day by the age of one year. The long-term use of these products does not help or even hinder the formation of the child’s taste and his/her chewing skills. 

### 4.10. Milky-Cereals Baby Food

The presence of added sugars in almost all the products of the milky-cereals group make it unnecessary to average out the sugars in the only two products that do not have any, but one cannot fail to point out the excessive amount of sugar in this type of baby food, which, in addition to reinforcing the liking of the sweet taste, can also then promote the development of dental caries. The WHO recommends that special flavorings such as vanillin should not be included in baby foods, as they change the taste and flavor of the natural food, depriving the infant of the right variety of tastes and flavors that allow them to develop healthy eating habits. 

### 4.11. Complete Baby Meals

Complete baby meals are products that, except for the two with added salt, might be useful at times when meal preparation for the infant is not possible. The age of 36 months as the final age for consumption is again unduly advanced. 

### 4.12. Dry Cereal Baby Food

In the dry cereal group, some products are enriched with iron, but the low number makes it almost inevitable to resort to other non-natural sources of this mineral such as milk formulas, while a greater number of enriched cereals would make it much easier to reach the PRI for iron. On the other hand, enrichment with vitamin B1 present in almost all products does not substantially improve the intake of the vitamin itself, which can also be taken from many other foods.

### 4.13. Biscuits

The use of biscuits is a legacy of times when diluted cow’s milk was currently used for infants. This was a good way to re-establish an appropriate energy and sugars intake, however, nowadays, makes no sense in CF since, when included in the formula, it increases its caloric intake in favor of carbohydrates and, in any case, given the sweet taste, does not promote taste development for flavors other than sweet itself. Generally, their texture is such that they very easily melt shortly after being put into the mouth. Therefore, if biscuits’ use is suggested by the stimulus, these products can provide chewing ability and hand-mouth movement coordination, which can also be achieved by offering products such as rusks or bread with no added sugars or, even better, by offering vegetables and/or fruit pieces cut into chunks. Even for biscuits, it is worth criticizing that they have too many added sugars and do not facilitate the development of healthy eating habits.

### 4.14. Spoon Baby Snacks

The fact that 92% of products have so much added sugar that 49% of the energy is given by total sugars also makes the use of these products risky for the development of dental caries and healthy eating habits. Another problem that arises with complex products is the possible confusion between the product name and the presence in the first place of an ingredient that is not the one by which the product is named (e.g., an “apple and yogurt snack” has as the top-ranking ingredient water, not apple) which can mislead the families in choosing the product; for example, in the spoon baby snack group, as many as 12 products that have fruit in the product name, have water as their first ingredient, and in a product named “apricot and yogurt snack”, the first ingredient is sugar, while yogurt is in the 5th place and apricot in the 14th place, with a percentage of 1.3. Such a composition is far from the product name and may induce choices made in good faith, but far from those that should be made for the health of children. In this group, the inconsistency between the product name and the first ingredient is present in twenty-nine products (39.7%). 

### 4.15. Drinking Baby Snacks (Fruit Juices or Purées, Yogurt)

Among the fruit purée-only products, 16 have added sugars from the same or other fruit concentrates than the main one. The others are yogurts with fruit and sweetened with added flavors particularly liked by children, such as vanilla and cocoa, which add nothing to children’s health. All these foods are high in sugars, especially fructose, which can predispose to the development of dental caries later on and can also represent a threat to the liver and kidneys, as well as being a co-inductor of metabolic syndrome [26,27]. 

In addition to nutritional harm, there is also harm posed by the formulation of some packaging. An increasing number of these products come in tetra packs with a spout and screw cap that the child can open to squeeze to suck/drink the liquid. This formulation does not help the development of chewing, encourages the consumption of sweet drinks instead of water, the formation of tooth decay due to the high intake of sugars, and finally, leaves the child alone at the time of intake, missing an opportunity for young children and parents to interact with each other during feeding [28].

### 4.16. Snacks-to-Bite

Although apparently useful and not in contrast with a normal diet, the mere presence of vitamins of group B in snacks-to-bite does not justify their use, especially since the quantities of these vitamins, and therefore the coverage quotas of DRV, are not declared on the label. Moreover, the richness of these products in sugars is not conducive to their healthy use.

### 4.17. Early Teething Snacks

Early teething snacks are the products that most respect the flavors and nutritional composition of the fruit from which they are composed. However, the mistake of recommending their use before six months remains. They could be used at times when natural fruit cannot be found, prepared, and consumed. However, the diction “early teething snacks” can mislead parents who might believe that these products are strongly advised at six months.

### 4.18. Baby Foods and WHO Recommendations

WHO has settled a number of rules for CBFs to be adequate from the nutritional point of view (Table 3) [12]. The WHO and Codex reference values [12,29] for the nutrients considered and with which to eventually compare those of the different CBFs can be found in the Appendix A.

#### 4.18.1. Energy

Since infants and toddlers require a high energy intake, but their stomach size can take only a small amount of foods, WHO recommends that CBFs in general should have at least more than 69 kcal (which is the energy provided by 100 mL of human milk) per 100 g of baby food ready-to-eat, whereas for dry products, for which water or other liquids have to be added to be edible, the energy content should be of 400 kcal/100 g of dry product to avoid the risk of malnutrition [12]. The recommendation on a minimum of energy content higher than 69 kcal/100 g of product could be correct if the infant only ate one type of food, but the diet at this age is composed of several foods that can and must integrate with each other to achieve maximum benefits. In addition, some natural foods, such as fruit, vegetables, and legumes also present on the market as CBFs and considered positive for health, have, within portion limits appropriate for this age, a natural energy content lower than that recommended but they are still necessary for the development of taste and for the content of some functional nutrients. On the other hand, many CBFs have an energy content (biscuits, baby cheese, some yogurt, etc.) well above 69 kcal/100 g and can cause excess of energy intake. Trying to walk the cutting edge between malnutrition and overnutrition, a solution could be to explain and also recommend a frequency of intake of different foods to families to balance the intake of energy and nutrients.

#### 4.18.2. Proteins

WHO recommends that protein content in baby foods has to be of high quality and of an adequate amount to allow infants’ optimal growth [12]. Existing Codex standards and EU regulations set minimum protein levels for different types of baby foods, according to the type of food [7,29].

The first factor that decides the minimum amount of protein that must be present in a baby food is the position that the protein source has in the name of the product. If the protein source is present as the first name in the product name (e.g., chicken purée with vegetables), the protein content must be at least 4 g per 100 kcal; if, on the contrary, the protein source is not in the first place (for example, vegetables purée with chicken), the quantity can go down to 3 g per 100 kcal. All the Italian CBFs are in line with this rule. However, in Europe, where malnutrition at such a young age is almost inexistent, this recommended amount of proteins in baby foods might be excessive, much more so if the infant is formula-fed, since a high protein content can reduce calcium intake at this age and also represent a risk factor for the development of obesity at later ages [30,31].

#### 4.18.3. Fats

The EU legislation and Codex standards recommend different upper limits for lipid content according to the type of food [27,28]. Briefly, as summarized by the WHO document, total fat should not exceed 4.5 g/100 kcal except for in some meat- fish- or cheese-based meals as they are naturally richer in fat content and with a fat content that can be <6 g per 100 kcal [12]. All the Italian CBFs that have been analyzed comply with these recommendations.

The EU legislation and Codex standards do not set any upper limits for saturated fatty acids. If WHO recommendations for children over the age of 2 years and adults can also be applied to infants, less than 10% of dietary energy should come from saturated fatty acids [7,29]. This means that the maximum amount of saturated fats allowed in CBF should be around 1.1 g per 100 kcal. Among Italian commercial baby cheese (3.1 g/100 kcal), pure meat purées (1.34 g/100 kcal), early teething meat (1.26 g/100 kcal), sauces (1.6 g/100 kcal), and spoon snacks (1.58 g/100 kcal), show a saturated fat content higher than 1.1 g/100 kcal. This aspect is generally less known by families and, as a consequence, pediatricians should inform the parents, in order to allow a better choice of baby foods for their children. There is little clarity on the labels about what types of fats are present in baby food. It would be desirable that the labels also indicate the quantity of essential fatty acids, and, above all, the quantity of omega 3 present, given their great importance in brain development [32]. 

#### 4.18.4. Sugar

In 2015, WHO published guidelines on sugar intakes for adults and children, reducing the recommended intake to <10% of total daily energy [15]. The reduction in intake is based on the evidence that an excess of sugar intake favors weight gain, dental caries and, in general, the development of NCDs.

Although the guidelines do not specifically speak of infants, the reduction of the presence of sugar in CBFs has a strong reason for being, due to the possible altered development of taste towards the sweet one, something that can persist even at later ages and in the possible formation of a favorable environment to the development of dental caries [33,34]. In the CBFs analyzed, several products exceed the 10% threshold, namely vegetables purées (22.7%), sauces (15.2%), Milky-cereal products (31.6%), complete baby meals (11.9%), biscuits (21.1%), spoon dairy snacks (38.4%), drinking snacks (fruit juice, yogurt) (67.7%), and early teething snacks (77.9%). Many of these CBFs are not essential for infant feeding and have no other reason than the habit of families to think that offering sweet foods to children does not have negative effects or even facilitate the growth of the infant because they are generally highly appreciated foods and consumed in greater quantities. Such products could and should be reformulated with the reduction of sugars present, if the baby food industry is truly interested in improving children’s health. 

#### 4.18.5. Sodium

According to WHO, maximum sodium content in all CBFs for infants and toddlers must not be higher than 50 mg per 100 g of product or 100 kcal. An exception is made for cheese purées and meals in which cheese is listed in the product name and where the dairy protein amount is higher than 2.2 g/100 kcal; in these cases, the limit is set at 100 mg/100 kcal and 100 mg/100 g of product [12]. The Italian CBFs are all compliant with this lower level recommended by WHO except for tomato sauces and baby cheese. The use of salt in CBFs is due in small part to the need to preserve the product, but, more often, it is due to the improvement of the flavor which, becoming more savory, is more appreciated not only by infants, but also by adults, who usually taste it before proposing it to the little ones. However, the great variability of sodium content between the various food groups, but also within the same group, makes it clear that it is possible to reduce the sodium content without storage problems. 

Data herein reported indicate that there is no need to add sugar or salt to infants’ diet. Similarly, sugar-sweetened beverages and juices should be restricted since it has been shown that their consumption positively correlated with adiposity gain. Finally, advertisements for high-fat, high-sugar foods should be limited and monitored to avoid demand for and consumption of those foods.

### 4.19. Texture

Food texture is an aspect that must be considered in the development of eating habits as many of the more protective foods (vegetables and fruit) have a greater consistency. Infants must be gently guided to learn how to manage foods with different textures to ensure that they eat a wide variety of foods and therefore a richer diet even at a later age. For this purpose, the CBFs are not of great help because the texture in purée clearly prevails, recommended even at an age in which they should no longer constitute the majority of the foods eaten.

### 4.20. The Italian Habits

Recently, one of the authors (R.F.) took part in a study that analyzed the intake of calories, macronutrients, fiber, minerals, and vitamins of 443 Italian children (6.4–131 months) using a three-day food record completed by their parents and evaluated by family pediatricians. The outcomes were evaluated in relation to the Italian DRV. The median protein consumption in g/kg per body weight surpassed the average requirement across all age categories, and the intake as a percentage of energy was above the reference range (>15 percent) throughout the 12–36-month period. The majority of the infants (82.3% of the population) consumed quantities of simple carbohydrates and saturated fats (69% of the population) above the limits of the Italian DRV, with low intake of fiber and polyunsaturated fats [35]. The findings of the research supplemented with data from the analysis of food questionnaires highlights the necessity for healthcare policies to be implemented as early as possible in order to improve nutrient intake throughout childhood, which may affect long-term health consequences.

## 5. Conclusions

Overall, it can be concluded that the time-space for which CBFs are recommended starts too early and ends too late. The protein content is adequate or even too high in some food. The amount of fats and their quality should be improved because of their special role in brain development, lowering the intake of saturated fats and raising the content in LC-PUFA. Again, too many baby foods are too rich in sugar. Its addition only serves to increase the sweet taste of the product, damaging the development of healthy eating habits, and favoring the loyalty to these products, thus facilitating their marketing. Thus, sweet snacks, in any form, including fruit as drink or purée, as well as cow’s milk with its derivatives, should not be on the market of baby food adequate to thirty-six months. The texture of too many products is that of a purée, and its use is recommended for too long of a time (thirty-six months), not facilitating and probably hindering the development of chewing abilities and of healthy eating habits. 

This article cannot establish or recommend whether CBF is qualitatively better or worse than home-prepared infant preparations, simply because there is no comprehensive and extensive information in different European countries on the nutritional composition of homemade foods. A study in Spain compared a full meal consisting of CBFs and a full meal consisting of homemade foods. Meals with homemade foods were found to be richer in protein and fiber and less energy-rich than those made up of CBFs sold in Spain [36].

We would like to stress, though, that all the CBFs taken into consideration comply with the Italian and European regulations, thus, one possible issue does not come from what the companies do, but from what the regulations allow them to do.

Knowing how important the nutrition of the first thousand days of life is, the research for the development of CBF preparations adapted to the nutritional and metabolic needs of children up to two years of age is of paramount importance. The search for the most appropriate composition of CBFs for the health and needs of infants must be a priority for the CBF industry. In this research field, pediatricians must play an important role, both as nutrition experts and as advocates of the health interests of children in comparison with the possible commercial economic interests of the industry. 

Therefore, in order to give children the highest nutritional standard, producers should be asked to improve the composition of CBFs. Furthermore, parents and caregivers in general must be informed and trained on the correct food preparation techniques for infants, so that there are neither nutritional excesses nor deficiencies that can negatively affect children’s growth and health. 

## 6. Strengths and Limitations

The strength of this article is to have evaluated CBFs available nation-wide in the Italian market with a careful analysis of their nutritional composition. The weakness is the lack of information on their costs. The variability of the price of the same product according to different sales channels (internet, pharmacy, supermarkets) and the frequent presence of offers and discounts makes it impossible to do a cost analysis of the products and the influence of this factor in purchasing choices.

## Figures and Tables

**Table 1 nutrients-14-03722-t001:** Suggested initial and final age for different types of baby food consumption. S.: Snacks.

Groups of CBFs	Fruit Purée	Vegetables Purée	Legumes Purée	Dairy Purée	Meat Only Purée	Meat or Fish with Vegetable Purée	Early Teething Meat	Sauces	Freeze-Dried Meat	Milky Cereal	Complete Meal	Biscuits	Dry Cereals		Baby Snacks			
													Dry Cereals	Baby Pasta	Spoon Baby S.	Drinking Baby S.	S. to Bite	Early Teething S.
**Total products** **number**	39	12	14	13	30	23	4	9	11	13	9	19	25	39	73	37	7	8
**Suggested initial age months**	4	4	4	4	4	4	8	8	4	4	6	4	4	4	4	4	4	4
**Products suggested at the initial age number** (%)	31 (79)	11 (91.7)	5 (35.7)	11 (85)	20 (67)	8 (39)	4 (100)	6 (66.7)	11 (100)	2 (18.2)	9 (100)	7 (37)	18 (72)	6 (15.4)	5 (6.8)	11 (30)	3 (37)	3 (37)
**Suggested consumption final age number** (%)	8 (20.5)	3 (25)	0	3 (23.1)	7 (23.3)	5 (21.7)	0	3 (33.3)	3 (27.3)	2 (15.4)	6 (66.7)	3 (15.8)	4 (16)	14 (35.9)	19 (26)	4 (10.8)	2 (28)	0

**Table 2 nutrients-14-03722-t002:** Nutritional analysis of commercial baby food available in Italy. * Values are expressed as M ± SD as g or mg/100 g or mL of product.

Products	Characteristics	N	Energy (kcal *)	Protein (g *)	Total Fat (g *)	Saturated Fat (g *)	Carbohydrates (g *)	Total Sugars (g *)	Fibre (g *)	Sodium (mg *)	Iron (mg *)	Calcium (mg *)
**Fruit purée**	**Overall**	39	62 ± 11	0.5 ± 0.2	0.3 ± 0.2	0.07 ± 0.1	14 ± 2	11 ± 2	1.5 ± 0.5	10 ± 7	-	-
	**Without added sugar**	23	63 ± 12	0.5 ± 0.2	0.3 ± 0.2	0.07 ± 0.1	14 ± 3	12 ± 2	1.7 ± 0.5	11 ± 8	-	-
	**With added sugar**	16	61 ± 10	0.4 ± 0.2	0.2 ± 0.2	0.07 ± 0.1	14 ± 2	10 ± 2	1.1 ± 0.2	9 ± 7	-	-
**Vegetables and legumes purée**	**Overall**	26	42 ± 8	1.8 ± 0.9	0.5 ± 0.4	0.07 ± 0.1	6 ± 2	2 ± 1	2 ± 1	19 ± 17	-	-
**Legumes purée**	**Overall**	14	45 ± 5	2 ± 0.6	0.5 ± 0.3	0.06 ± 0.1	6 ± 1	2 ± 1	3 ± 1	12 ± 9	-	-
**Cheese purée**	**Overall**	13	112 ± 22	9 ± 3	5 ± 2	3 ± 2	7 ± 1	1 ± 1	-	236 ± 130	-	258 ± 165
	**Without added salt**	5	106 ± 13	8 ± 0.8	5 ± 1	3 ± 1	8 ± 1	1 ± 1	-	229 ± 114	-	-
	**With added salt**	8	116 ± 26	9 ± 3	6 ± 2	4 ± 2	7 ± 2	1 ± 2	-	2 ± 147	-	286 ± 158
	**Without added calcium**	13	115 ± 22	9 ± 3	6 ± 2	4 ± 1	7 ± 1	1 ± 1	-	268 ± 113	-	322 ± 136
	**With added calcium**	2	97 ± 6	7 ± 1.4	4 ± 0.4	1 ± 0.4	9 ± 2	4 ± 2	-	60 ± 28	-	66 ± 2
**Meat purée**	**Overall**	30	79 ± 5	6 ± 0.5	3 ± 1	1 ± 0.3	7 ± 1	0 ± 0	-	37 ± 32	0.4 ± 0.2	-
**Meat or fish purée with vegetables**	**Overall**	23	66 ± 8	4 ± 0.7	2 ± 1	0.4 ± 0.3	8 ± 1	0 ± 0	-	33 ± 26	0.4 ± 0.4	-
**Meat (early teething)**	**Overall**	4	82 ± 2	6 ± 0	3 ± 0.2	1 ± 0.1	7 ± 0	0 ± 0	-	26 ± 2	0.4 ± 0.2	-
**Sauces**	**Overall**	9	64 ± 8	3 ± 1	2 ± 0.5	1 ± 1	7 ± 1	2 ± 1	-	76 ± 41	-	-
	**Without added salt**	3	68 ± 10	3 ± 2	3 ± 1	1 ± 1	7 ± 1	2 ± 1	-	30 ± 19	-	-
	**With added salt**	6	62 ± 6	3 ± 1	2 ± 0.3	1 ± 0.4	7 ± 1	2 ± 1	-	99 ± 27	-	-
**Freeze-dried meat**	**Overall**	11	440 ± 22	48 ± 4	12 ± 4	5 ± 2	34 ± 3	0 ± 0	-	119 ± 27	-	-
**Milky cereal**	**Overall**	13	413 ± 18	3 ± 3	9 ± 3	4 ± 1	69 ± 6	33 ± 7	-	119 ± 46	6 ± 1.8	384 ± 82
**Complete meal**	**Overall**	9	76 ± 20	3 ± 0.6	2 ± 0	0.4 ± 0.2	9 ± 1	2 ± 1	-	37 ± 22	-	-
**Cereal meal**	**Overall**	64	361 ± 20	10 ± 2	2 ± 1	0.3 ± 0.1	76 ± 5	4 ± 6	-	13 ± 5	9.5 ± 0.9	231 ± 118
	**Without added iron**	49	363 ± 22	10 ± 2	2 ± 1	0.4 ± 0.2	76 ± 5	3 ± 4	-	14 ± 6	-	264 ± 156
	**With added iron**	15	355 ± 15	10 ± 0.7	1 ± 0.2	0.3 ± 0.1	74 ± 4	8 ± 8	-	13 ± 1	9.5 ± 0.9	195 ± 10
	**Without added calcium**	45	362 ± 23	10 ± 2	2 ± 1	0.3 ± 0.2	76 ± 5	3 ± 4	-	14 ± 5	-	184 ± 8
	**With added calcium**	19	359 ± 15	10 ± 1	1 ± 0.3	0.3 ± 0.1	75 ± 5	8 ± 8	-	12 ± 5	9.6 ± 0.9	265 ± 145
**Cereal cream**	**Overall**	25	381 ± 8	9 ± 2	2 ± 1	0.3 ± 0.3	80 ± 6	6 ± 8	-	16 ± 5	7.8 ± 0.3	376 ± 200
	**Without added iron**	22	380 ± 8	9 ± 3	2 ± 2	0.3 ± 0.3	79 ± 6	4 ± 6	-	16 ± 6	-	525 ± 104
	**With added iron**	3	383 ± 3	9 ± 1	1 ± 0.5	0.3 ± 0.1	82 ± 2	23 ± 4	-	16 ± 0	7.8 ± 0.3	177 ± 6
	**Without added calcium**	18	382 ± 8	10 ± 3	2 ± 2	0.3 ± 0.3	80 ± 7	3 ± 6	-	17 ± 4	-	-
	**With added calcium**	7	377 ± 7	9 ± 2	2 ± 0.5	0.4 ± 0.2	80 ± 4	13 ± 10	-	12 ± 8	7.8 ± 0.3	376 ± 200
**Baby pasta**	**Overall**	39	348 ± 16	10 ± 1	1 ± 0.2	0.3 ± 0.1	73 ± 2	3 ± 3	-	12 ± 43	10 ± 0	191 ± 10
	**Without added iron**	27	348 ± 19	10 ± 1	1 ± 0.2	0.4 ± 0.1	73 ± 2	3 ± 2	-	12± 5	-	183 ± 7
	**With added iron**	12	348 ± 1	10 ± 0.1	1 ± 0	0.3 ± 0	72 ± 1	4 ± 3	-	12 ± 0	10 ± 0	200 ± 0
	**Without added calcium**	27	348 ± 19	10 ± 1	1 ± 0	0.4 ± 0.1	73 ± 2	3 ± 2	-	12 ± 5	-	183, ± 7
	**With added calcium**	12	348 ± 2	10 ± 0.1	1 ± 0	0.3 ± 0	72 ± 1	4 ± 3	-	12 ± 0	10 ± 0	200 ± 0
**Biscuits**	**Overall**	19	425 ± 14	8 ± 2	10 ± 2	3 ± 2	75 ± 2	22 ± 4	-	132 ± 57	5.6 ± 0.6	296 ± 30
	**Without gluten**	3	433 ± 12	6 ± 2	11 ± 3	4 ± 3	77 ± 4	23 ± 2	-	113 ± 95	5.7 ± 0.6	273 ± 35
	**With gluten**	16	424 ± 14	9 ± 1	9 ± 2	3 ± 2	75 ± 2	22 ± 4	-	135 ± 51	5.6 ± 0.6	306 ± 23
**Spoon baby snacks**	**Overall**	73	94 ± 38	2 ± 1	2 ± 2	1 ± 1	16 ± 5	11 ± 4	-	49 ± 152	-	-
	**Without added sugar**	6	59 ± 8	0.6 ± 0.22	0.3 ± 0.2	0.1 ± 0.1	13 ± 2	10 ± 1	-	7 ± 6	-	-
	**With added sugar**	67	97 ± 38	2 ± 1	3 ± 2	2 ± 1	16 ± 5	12 ± 4	-	53 ± 158	-	-
**Drinking baby snacks**	**Overall**	37	69 ± 20	1 ± 1	1 ± 1	0.6 ± 0.8	13 ± 3	12 ± 2	-	15 ± 14	-	-
	**Without added sugar**	9	62 ± 9	0.6 ± 0.4	0.4 ± 0.4	0.2 ± 0.3	13 ± 2	11 ± 2	-	6 ± 8	-	-
	**With added sugar**	28	71 ± 22	1 ± 1	1 ± 1	0.6 ± 0.9	14 ± 3	12 ± 3	-	18 ± 14	-	-
**Snacks-to-bite**	**Overall**	7	408 ± 24	8 ± 2	8 ± 9	3 ± 5	76 ± 17	9 ± 13	-	75 ± 69	-	-
**Early teething snacks**	**Overall**	8	61 ± 8	0.5 ± 0.1	0.3 ± 0.2	0.01 ± 0.01	13 ± 2	12 ± 2	-	15 ± 3	-	-

**Table 3 nutrients-14-03722-t003:** Energy and nutrients content of baby foods per 100 kcal or as Energy% 100 kcal [12].

Baby Food Groups	Baby Food Subgroups	Energy/100 g of Product (kcal) °	Protein g/100 kcal ^	Saturated Fats g/100 kcal #	Total sugar %En/100 kcal +	Sodium mg/100 kcal *
**Fruit purée**		**62.4**	0.78	0.11	**18**	17
**Vegetables purée**		**38.8**	2.6	0.2	5.7	71.7
**Legumes purée**		**45.4**	5.6	0.1	**22.7**	26.7
**Baby cheese purée**		112.2	7.74	**3.1**	3.48	**210**
**Meat-only purée**		79.3	7.9	**1.34**	0.02	46
**Meat or fish with vegetables purée**		**66**	5.9	0.06	1.82	49.7
**Early teething meat**		81.8	7.58	**1.26**	0	31.2
**Tomato sauces**		**63.8**	4.8	**1.6**	**15.2**	118.7
**Freeze-dried meat baby food**		440	10.9	1.04	0	27.9
**Milky-cereal baby food**		413	3.1	0.88	**31.6**	29.9
**Complete baby meals**		75.7	3.6	0.47	**11.9**	48.8
**Dry cereals baby food**	Dry cereals	380.6	2.48	0	6.43	4.1
	Baby pasta	348.4	2.91	0	3.9	0
**Biscuits**		425.2	1.97	0.75	**21.1**	30.9
**Baby snacks**	Spoon baby snacks	**37.7**	2.35	**1.58**	**38.4**	52.5
	Drinking baby snack (fruit juice, yogurt)	68.8	1.58	0.85	**67.7**	22
	Snack-to-bite	408.3	2	0.87	9	18
	Early teething snacks	**60.6**	0.8	0	**77.9**	24.7

Values lower or higher than recommended in bold. ° Minimum value >69 kcal/100 g of product (kcal). ^ variable according to the type of food and label. # <1.1 g/100 kcal. + recommended <10% energy/100 kcal. * upper limits for sodium content <50 mg per 100 kcal in cereals; 100 mg per 100 kcal for fruit/vegetable purees and fruit/vegetable with cereal products, meat-, fish- or cheese-based meals, juices/drinks, sweet or savory snacks.

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
