# Peer review of "Nutritional Assessment of Baby Food Available in Italy"

_nutrients, 2022, doi:10.3390/nu14183722_

Round 1

Reviewer 1 Report

The abstract is missing the methodology explanation. And the Keyword weaning seems to be a bit out of context;

In Materials and Methods the authors do not mention what how, when and where the articles were searched. How many were first pooled out and how many were excluded and why.

But in overall, the content is very elucidative and pertinent for nutrition in pediatric field

Reviewer 2 Report

Manuscript: Nutritional and safety assessment of baby food available in Italy 2

I would like to thank the editors for providing me opportunity to review this manuscript.

Despite that the content of the manuscript is important, however the data should be well structured to show the pediatric health risk through the consumption of these products. I guess, through this outcome, your paper can be publishable.

For instance, I would suggest to include the cutoff of the regulations (Codex or EFSA or etc..) next to each value of the variables tested. In this way you can compare how much the content exceed the regulations or even you can conclude how much this product can contribute to the RDA per day in 1000 Kcal.

In this case, you can compare these values and include a p-value showing the difference.

If the variables tested exceed the RDA, in this case there are many cumulative pediatric risk and this can threaten the Italian children.

I suggest in the discussion part to include the reasons for adding higher amounts of sugar for example to pureed baby food. Similarly in all other paragraphs. For example, in lines 509, why don’t you include RDA values and comparison as discussed previously. Also, please state the reason for adding Na to these foods. What are the forms of Na added in this kind of foods…

Line 535: I am not convinced in adding this paragraph. Nowadays, we don’t talk again about the terminology of food safety, but about health risk assessment showing exposure of children to contaminants through the consumption of foods. I don’t agree with you concerning the terminology “Windows of vulnerability of infants.: please change.

There is no need to add these info inside Table 4. Please add these data inside a separate paragraph and remove Table 4.

Also, In you title, you are announcing assessment of food safety of baby food products. Where are the assessment variables?

I suggest to keep the manuscript on showing nutrition content of baby food products sold in Italy. It is better to change the title and delete the part related to food safety.

This manuscript cannot be accepted without taking into consideration these changes.

Reviewer 3 Report

In the manuscript, entitled “Nutritional and safety assessment of baby food available in Italy” sent for the potential publication in Nutrients, a research analysis of nutritional composition of baby food available in Italy is presented. I am of opinion, that in the present form, the article is not suitable for the publication in Nutrients and substantial revision has to be made before re-evaluation for potential acceptance.

My comments:

1.       In the abstract, the results have to be presented in more detail.

2.       The course of data gathering should be described more specifically.

3.    Some comparison with available baby food in other countries should be made as well as with the expert recommendations.

4.  A more detailed texture of different foods as well as some other micronutrients, e.g. zinc, should be included in the review.

5.       It has to be clearly stated what new knowledge the article adds to the field of nutrient composition of baby food.

6.       Clinical importance of baby food composition research has to be discussed.

7.    Basic straightforward recommendatons based on the research regarding available food products have to be prepared.

Reviewer 4 Report

This manuscript discusses the formulation of several different types of commercial baby food and their increased consumption in Italy. The aim of the study was to perform an up-to-date analysis of the nutritional composition of commercial baby food available in the Italian market. As a result macronutrient content is compared to guidelines and it is found that often the protein content is sufficient or even high, lipids could be of better quality and finally sugar having too high a content in many cases. 

As a reviewer I have some general comments:

* The manuscript is classified as a Review article (first page) but contains information gathered by the methods described. I find it more appropriate to call the manuscript an article. 

*English is ok, however the abstract must be improved, eg the placement of "low" on row30 in the end of the sentence, abbreviations used withour explanation and the word " could" on row 29 is likely better described as "must" if the quality is so important for brain development

*Data are gathered over different years but no trend is discussed looking at the differences between the years. This could be a valuable addition to the manuscript.

Specific comments:

*Row 61 error of aim? "confuse family choice" I guess you mean: "can confuse familys to make bad choices" 

*In the method section -  please mention how you handled ambiguous data (perhaps not a big issue when only looking at macronutrients) but often you see mmol/mg or IE as unit and sometimes the content is expressed as dry matter/in solution, containg a salt or hydrated. Did you encounter these problems and in such case how did you handle it?

*Table 1 - different spelling of puree in the paper/table. Typ0 for freeze

*If suggested initial age is stated in the table - why is not the sugested final age mentioned?

*Table 2 - caption. You must mention the unit that the content is presented for- Per 100g /100ml or serving?

*Table 2 - Many readers are likely unfamiliar with the meaning of "kcal%g" please explain. Also mg% is seen on row 148

*Table 2 - some figures contain too many figures - Are they significant? Please using rounding . Eg 116,13 is five (!) significant figures - food can never be so exact!

*Table 2 is large - can you indicate when a product group is broken into two pages that it continues, eg "Cereal Meal (continued)" on page2

*Row 215 "addiction" should read "addition"?

*Row 220 "According to the Italian regulations..." I suggest you add "are not allowed" As the sentence reads in its current way it is ambiguous. 

*Row 279 - linguistic aim . A product cannot release information. It is the manufacturer of the product that releases information. 

*Row 344 should prompt (no s)

*Row 371 the abbreviation DRV is mentioned - please explain together with PRI - likely in the introduction. DRV returns on row 432 but without the abbreviation...

*Table 3 - "values lower or higher are bold" - why is not the >400 kcal energy values bold?

*Row 482 "complain" - should read comply?

*Row 492 needs a reference to the text on brain development

*Row 509 - "shouldn't" is perhaps not used in written scientific papers?

*Row 510 - needs a space "100 kcalories"

*On row 501 decimal comma is used but on row 529 decimal point is used. To be the same way throughout the manuscript. 

Round 2

Reviewer 2 Report

comments are taken into consideration